# Systematic bias of Tibetan Plateau snow cover in subseasonal-to-seasonal models

Wenkai Li[1], Shuzhen Hu[1], Pang-chi Hsu[1], Weidong Guo[2], Jiangfeng Wei[1]

[1]Key Laboratory of Meteorological Disaster, Ministry of Education (KLME)/Joint International Research Laboratory of Climate and Environment Change (ILCEC)/Collaborative Innovation Center on Forecast and Evaluation of Meteorological Disasters (CIC-FEMD), Nanjing University of Information Science & Technology, Nanjing, 210044, China
[2]Institute for Climate and Global Change Research, School of Atmospheric Sciences, Nanjing University, Nanjing, 210023, China

*Correspondence to*: Wenkai Li (wenkai@nuist.edu.cn)

**Abstract.** Accurate subseasonal-to-seasonal (S2S) atmospheric forecasts and hydrological forecasts have considerable socioeconomic value. This study conducts a multimodel comparison of the Tibetan Plateau snow cover (TPSC) prediction skill using three models (ECMWF, NCEP and CMA) selected from the S2S project database to understand their performance in capturing TPSC variability during wintertime. S2S models can skilfully forecast TPSC within a lead time of 2 weeks but show limited skill beyond 3 weeks. Compared with the observational snow cover analysis, all three models tend to overestimate the area of TPSC. Another remarkable issue regarding the TPSC forecast is the increasing TPSC with forecast lead time, which further increases the systematic positive biases of TPSC in the S2S models at longer forecast lead times. All three S2S models consistently exaggerate the precipitation over the Tibetan Plateau. The exaggeration of precipitation is prominent and always exists throughout the model integration. Systematic bias of TPSC therefore occurs and accumulates with the model integration time. Such systematic biases of TPSC influence the forecasted surface air temperature in the S2S models. The surface air temperature over the Tibetan Plateau becomes colder with increasing forecast lead time in the S2S models. Numerical experiments further confirm the causality. These incorrect cooling shifts over the Tibetan Plateau are caused by the systematic biases of TPSC.

## 1 Introduction

Anomalous weather- and climate-related natural disasters are among the most common disasters and are associated with severe socioeconomic consequences. Reliable forecasts of such weather and climate anomalies with sufficient lead time have significant benefits for decision-makers (White et al., 2017). Traditionally, weather forecasts cover a time range of up to 2 weeks, while climate forecasts begin at the seasonal timescale and extend outward. Demands are growing rapidly in operational forecasts in the subseasonal-to-seasonal (S2S) range (from two weeks to a season). The primary basis for longer lead forecasts beyond 2 weeks is the interaction of the atmosphere with other, more slowly varying earth system components, such as the ocean or land, that evolve over timescales of weeks and months, rather than days as in the atmosphere (Mariotti et al., 2018).

Land–atmosphere coupling is one of the key physical processes for S2S prediction but is not well simulated and may reduce S2S prediction skill (Robertson et al., 2014; Dirmeyer et al., 2019).

Snow cover is a crucial component in both the climate system and the cryosphere. The radiative and thermal properties of snow cover significantly influence the ground thermal regime (Zhang, 2005). As the lower boundary condition of the atmosphere, snow cover forces the regional and global atmosphere and can serve as an indicator of the atmosphere (Barnett et al., 1989; Bamzai and Shukla, 1999; Wu and Kirtman, 2007; Henderson et al., 2018). Snow cover can vary rapidly within a season over discontinuous or sporadic permafrost zones (Wang et al., 2015; Suriano and Leathers, 2018; Song et al., 2019; Li et al., 2020a) and rapidly influence the atmosphere (Clark and Serreze, 2000; Zhang et al., 2019). Snow cover may provide a potential source of S2S predictability via its variability and atmospheric effects at the subseasonal time scale (F. Li et al., 2019; Diro and Lin, 2020).

The Tibetan Plateau is the highest plateau in the world and is known as the "third pole". Due to its high elevation and cold climate, the Tibetan Plateau has much more snow cover than the other regions at the same latitude. Tibetan Plateau snow cover (TPSC) is a key component of the climate system. TPSC influences land surface thermal conditions (Chen et al., 2017; Li et al., 2018) and thus influences atmospheric circulations and monsoons over Asia and beyond (Wu and Qian, 2003; Lin and Wu, 2011; Xiao and Duan, 2016; Wang et al., 2017; You et al., 2020). TPSC shows variations at multiple time scales, including the subseasonal scale (Li et al. 2016; Song and Wu, 2019; Li et al., 2020a). The subseasonal variations in TPSC influence the atmosphere over East Asia (Li et al., 2018; Li et al., 2020b). A better TPSC simulation and forecast may favour a better forecast for weather and climate at the S2S time scale.

Snow cover also affects the hydrologic cycle. The accumulation of precipitation in the form of snow and its release through snowmelt runoff is an important component of the hydrologic cycle (Jeelani et al., 2012; Fayad et al., 2017). TPSC plays an important role in hydrological systems, providing a reservoir of water and acting as a buffer that controls river discharge. Rivers including the Yangtze River, Yellow River, Yarlung Zangbo River and Mekong River have headwaters over the Tibetan Plateau. Studies on the variability in TPSC are critical for water management in downstream regions (Immerzeel et al., 2009; Zhang et al., 2012; Zhang et al., 2013). Skilful predictions of TPSC with sufficient lead time are thus of great societal importance for hydrologic prediction.

Since the implementation of the S2S prediction project database (Vitart et al., 2016), many studies have evaluated the skill of S2S models for atmospheric elements and variables, such as the Madden–Julian Oscillation (Vitart, 2017), surface air temperature (Yang et al., 2018; Wulff and Domeisen, 2019), and precipitation (de Andrade et al., 2019). Some works also focus on the skill of S2S models for hydrological elements (Wei Li et al., 2019; Schmitt Quedi and Mainardi Fan, 2020). However, we still know little about the skill of S2S models for TPSC. Understanding the forecasting skills of the S2S model on the TPSC is the first step to applying the S2S model to hydrological forecasts over the Tibetan Plateau. Moreover, considering the influence of TPSC on the atmosphere, clarifying the issue of the S2S model for TPSC helps improve the ability of the S2S model for atmospheric forecasting.

This study conducts a multimodel comparison of the TPSC prediction skill using selected models from the S2S project database to learn about their performance in capturing TPSC variability. Our main goal is to use the state-of-the-art S2S prediction systems of these operational centres to demonstrate why models exhibit systematic biases of TPSC and whether such systematic biases influence the regional air temperature forecasted in S2S models. The remainder of this paper is organized as follows. Details on the data set and method used in this study are described in Section 2. The systematic bias of TPSC in S2S models and its effect on local temperature during wintertime are presented in Section 3 and Section 4, respectively. The conclusions and a discussion are presented in Section 5.

## 2 Data and method

### 2.1 S2S forecast models

The reforecasts considered for this study are taken from three operational forecast systems that are part of the S2S project database: the European Centre for Medium Range Weather Forecasts (ECMWF), the US National Centers for Environmental Prediction (NCEP), and the China Meteorological Administration (CMA). These models share a common reforecast period of 1999–2010 with a reforecast initialized frequency that is equal to or greater than once a week. This study only used reforecasts produced by the control forecast (using a single unperturbed initial condition). Details of the S2S database can be found in Vitart et al. (2016). Daily reforecast data were averaged for each 7-day period starting every 1 January to create a total of 52 weeks per year (December 31 was excluded). The reforecasts that initialized on the first day of these weeks were selected. Forecast lead times were defined here as 1 week (1–7 days), 2 weeks (8–14 days), 3 weeks (15–21 days), 4 weeks (22–28 days), and 5 weeks (29–35 days).

For the ECMWF model, the reforecasts initialization is based on ERA-Interim and ERA-Interim/Land datasets. The daily Interactive Multisensor Snow and Ice Mapping System (IMS) snow-cover product has been used to constrain the ERA-Interim snow analysis (Dee et al., 2011). The NCEP model also initialize realistic snow in the forecasts. The snow initialization comes from the Climate Forecast System Reanalysis snow analysis using IMS and the Air Force Weather Agency snow depth analysis. Snow in the CMA model was not directly initialized in the forecasts. The initial conditions of the snow in the CMA model are from a balanced state produced by long-term air-sea initialization integration. See the details on snow initialization in the S2S models at https://confluence.ecmwf.int/display/S2S/Models.

The land surface models used for ECMWF, NCEP and CMA are the Hydrology Tiled ECMWF Scheme for Surface Exchanges over Land (HTESSEL; Balsamo et al., 2009), Noah (Ek et al., 2003) and BCC_AVIM2 (Wu et al., 2014), respectively. All these land surface models contain snow schemes. According to the snow scheme in each land surface model, we obtain the snow cover fraction, which is a diagnostic variable in this study.

The snow cover fraction ($f_{snow}$) in the ECMWF model is parameterized as follows:

$$f_{snow} = min[\ 1,\ S/(0.1 \times \rho)\ ] \tag{1}$$

where *min* indicates the minimum function, *S* is the snow water equivalent (unit is kg m$^{-2}$), and $\rho$ is the snow density (unit is kg m$^{-3}$) (Dutra et al., 2010).

The $f_{snow}$ in the NCEP model is parameterized as follows:

$$f_{snow} = min[\ 1, 1 - (e^{-0.001 \times 2.6 \times S/SNUP} - 0.001 \times S/SNUP \times e^{-2.6})\ ] \tag{2}$$

where *e* is the natural logarithm, and *SNUP* is the vegetation parameter, which indicates the threshold snow depth (in water
equivalent m) that implies 100% snow cover (Koren et al., 1999; Ek et al., 2003). The *SNUP* ranges from 0.01 to 0.08 for different vegetation types. Details on the Noah code and vegetation parameters can be accessed in https://ral.ucar.edu/solutions/products/unified-noah-lsm.

The $f_{snow}$ in the CMA model is parameterized as follows:

$$f_{snow} = min[\ 1, 1.77 \times d/(d + 10.6)\ ] \tag{3}$$

where *d* is the snow depth (unit is cm), which is calculated from the snow water equivalent and snow density (Wu and Wu, 2004).

The surface air temperature (SAT) in these S2S models is also used. All variables are at a 1°×1° horizontal spatial resolution.

## 2.2 Validation data and method

The Tibetan Plateau area of focus in this study is the region within 26–41°N and 70–105°E at an altitude of greater than 3000 m (Fig. 1). Although the Tibetan Plateau is located over middle latitudes, the area is cold due to high altitude, especially in boreal winter. This study focuses on TPSC during wintertime. Here, each winter contains 17 weeks, covering from the 45th week (November 5–11) in one year to the 9th week (February 26–March 4) in the following year. This study spans 11 winters (from 1999/2000 to 2009/2010).

The reforecasts in the S2S models are verified against observational daily snow cover and SAT in the reanalysis. Observational daily snow cover data are obtained at a 24 km resolution from the Interactive Multisensor Snow and Ice Mapping System (IMS) snow cover analysis (Helfrich et al., 2007) provided by the National Oceanic and Atmospheric Administration. The IMS examines satellite images and other sources of data on snow cover and generates maps of snow cover distribution. The IMS analysis over the Tibetan Plateau corresponds well with ground-based measurements and can capture the general
subseasonal variability in TPSC (Yang et al., 2015; Li et al., 2018). The original 24-km resolution IMS analysis is interpolated into the 1°×1° grid of the S2S models. IMS provides binary snow cover information: it has the value of 1 if more than 50% of the 24-km pixel is covered by snow; otherwise, it is 0 (snow free). Orsolini et al. (2019) aggregated the original IMS product to a lower resolution rectilinear grid. They counted the number of pixels with a value of 1 in a grid box; assuming that they have 100% cover gave the high estimate, and assuming that they represent 50% cover gave the low estimate. These two
estimates provide a range of values, which reflects the uncertainty inherent to aggregating the 24-km binary data, e.g., a value of 1 in a pixel means a 50% to 100% snow coverage. Here, we used a method similar to Orsolini et al. (2019) to interpolate the original IMS product into the 1°×1° grid of S2S products, but we further averaged these two estimates. Daily SATs at a

1°×1° resolution are obtained from ERA-Interim reanalysis (Dee et al., 2011). These data range from 1 January 1999 to 30 December 2010. S2S reforecasts are compared with the observations and reanalysis for the same calendar date.

Two precipitation datasets, the Global Precipitation Analysis Products of the Global Precipitation Climatology Centre (GPCC; Schneider et al., 2011) and the Tropical Rainfall Measuring Mission (TRMM; Huffman et al., 2007), are used to evaluate the wintertime mean precipitation. The GPCC precipitation dataset is from rain-gauges built that were GTS-based. The TRMM precipitation dataset is based on satellite observations. The precipitation used in this study spans 11 winters (from 1999/2000 to 2009/2010).

To quantify the forecast ability of S2S models, three common statistical measures, i.e., the temporal correlation coefficient (TCC), the root-mean-square error (RMSE) and the mean bias, are calculated in this study. A composite analysis is performed to investigate the different performances on predicting the snow cover for increasing cases and decreasing cases (details are described in Section 3.2).

## 2.3 Numerical model and experimental design

To reveal the causality of the systematic bias of the TPSC-induced regional SAT bias, numerical experiments are performed. Numerical experiments are performed using the Advanced Weather Research and Forecasting Model (WRF-ARW, version 4.1.3), which was developed by the National Center for Atmospheric Research (NCAR). WRF-ARW has been applied to climate research, including studies of land-atmosphere interactions. The land surface parameterization scheme used in this study is the Noah land surface model (Ek et al., 2003). Important physics options include the WRF single-moment 6-class

microphysics scheme (Hong and Lim, 2006), the NCAR Community Atmosphere Model (CAM 3.0) spectral-band shortwave and longwave radiation schemes (Collins et al., 2006), the Yonsei University planetary boundary layer scheme (Hong et al., 2006), and the Kain–Fritsch convective parameterization scheme (Kain, 2004). The WRF is driven by atmospheric and surface forcing data extracted from the National Centers for Environmental Prediction (NCEP) FNL Operational Model Global Tropospheric Analyses. The simulation domain is in a cylindrical Equidistant projection with a horizontal resolution of 1°×1°

and located within 5–65°N and 40–170°E (as shown in Fig. 1) and without nesting. There are 41 levels in the vertical direction.

Two ensemble experiments are performed: control (CTL) runs and sensitive experimental (EXP) runs. All these runs have the same initial times as the forecasts in the S2S models that we used in this study for each winter. But the experiments were run for 20 winters (from 2000/2001 to 2019/2020), and both runs contain 340 cases. Each member ran continuously for 22 days. The first day in each run is for spin-up, and the results are discarded. The CTL runs are integrated freely without any

modification. Because both the NCEP S2S model and our numerical experiment use Noah as the land surface model, the TPSCs in CTL runs are expected to show unreal increases with integration time, which is similar to that in the NCEP S2S model (will be revealed in Section 3). The EXP run is designed to eliminate such bias in TPSC. The FNL analyses are from the Global Data Assimilation System (GDAS), which continuously collects observational data from the GTS and other sources for many analyses. GDAS incorporates daily snow data from IMS analyses and the Air Force Weather Agency Snow Depth

Analysis Model. We replace the forecasted TPSC in the WRF model with TPSC in the FNL analyses every 6 hours. Because

FNL analyses assimilate the observed TPSC, the TPSC in the EXP run is expected to show a small bias that increases with integration time. We averaged all 340 cases in CTL runs and EXP runs respectively. Ensemble mean results between CTL and EXP are compared with each other.

## 3 Tibetan Plateau snow cover in the S2S forecast models

### 3.1 Increasing Tibetan Plateau snow cover with forecast lead time

Before we present the systematic bias of TPSC in the S2S models, the overall forecast skills of TPSC is evaluated. Here, we focus on the variation in snow-covered area over the entire Tibetan Plateau, which can be measured by a TPSC index. The TPSC index represents the percentage of grid points covered by snow in the analysis or models over the entire Tibetan Plateau. The unit of the TPSC index is %. The prediction skill of the TPSC index has been investigated through the TCC and RMSE between the TPSC index in the predictions and that in the observations during wintertime (Fig. 2). A skilful prediction is generally defined as a TCC greater than 0.5. All three models show good prediction skills at lead times of 1–2 weeks with a TCC greater than 0.5 (Fig. 2a). At lead times of 1–2 weeks, the TCC for the ECMWF model is largest among the three models. The NCEP model has the lowest TCC among the three models at a lead time of 1 week. However, the TCC for NCEP falls the most slowly at lead times of 2 weeks or more. The NCEP model has a larger TCC than the CMA model at lead times of 2 weeks or more. The TCC values decrease with the increase in the forecast lead time and decline below 0.5 at and after lead times of 3 weeks for all three models. RMSEs increase with the forecast lead time (Fig. 2b). The RMSE for ECMWF is the smallest among the three models. Additionally, CMA has the largest RMSEs. These results indicate that the S2S models can skilfully forecast TPSC variations within a lead time of 2 weeks during wintertime but show limited skill at a lead time of 3 weeks or more.

The above results also indicate that the ECMWF model is shown to have a better TPSC forecasting skill than the other two models. Even so, the ECMWF model shows nonnegligible RMSEs with a TPSC index of more than 15% (Fig. 2b). The other two models, especially the CMA model, show even more significant RMSEs up to more than 25%. These large errors in the forecasting of the TPSC are induced by systematic bias of the TPSC, as shown by the following. The multiyear wintertime mean biases of the TPSC index in forecasts against that in the IMS snow cover analysis for all three models show positive values, which indicates that all of the models tend to overestimate the TPSC during winter (Fig. 3a). The TPSC index in the ECMWF is higher than the observed TPSC index by approximately 20–30%. NCEP has a larger TPSC index than that in the observation by approximately 5–20%. The CMA shows largest biases of approximately 25–40%.

Another remarkable issue regarding the forecast of TPSC is the increasing TPSC with forecast lead time, which further increases the overestimation of TPSC in models at longer forecast lead times. These increasing biases can be detected from the multiyear winter mean biases (Fig. 3a). To highlight such increasing biases, we further present differences in the multiyear winter biases for the TPSC index between forecasts for leads of 2–5 weeks and forecasts for leads of 1 week in three modes (Fig. 3b). Such differences are obtained by subtracting the multiyear winter mean of TPSC index at a lead time of 1 week from

that at forecast lead times of 2–5 weeks. The differences in the three models show common features: the differences in all three models are all positive and increase with increasing forecast lead time. The positive biases of TPSC with the longest forecast lead time (5 weeks) are largest among all forecasts. The increases in the differences in the ECMWF model are the smallest, while the CMA model has the largest increases in the differences. Taking the differences between the forecasts with a lead of 4 weeks and the forecasts with a lead of 1 week as an example, the spatial patterns of these increases in the biases in the three models show some similarities (Fig. 4). Although the spatial patterns of the differences in the three models show some small discrepancies, the differences are mainly positive in the three models, especially over parts of central and eastern Tibetan Plateau. These indicate that the increasing TPSC with the forecasting lead time occur at a regional scale.

## 3.2 Snow cover accumulation versus dissipation

The intraseasonal variability in TPSC leads to obvious rapid variations in TPSC with a period shorter than a season, making TPSC exhibit a distinct lack of persistence within one season (Li et al., 2020a). Both accumulation and dissipation of snow cover occur within a season over the Tibetan Plateau. The increase in TPSC with forecast lead time in the models may be induced by overestimation of snow cover accumulation or underestimation of snow cover dissipation. To support this hypothesis, we analysed the frequency of weekly TPSC accumulation and dissipation in the observation and forecast models in winter (Table 1). Here, the increasing (decreasing) weeks means that the TPSC index is greater (less) than that in the preceding week. The TPSC indexes in the S2S models are compared with the TPSC index in the preceding week, which are initialized at the same time, but with different forecast lead times.

The proportions of increasing and decreasing weeks in the observations are 50.3% and 49.7%, respectively, which is fairly even (Table 1). However, this kind of balance does not exist in the models. In the models, the proportion of increasing weeks is mostly more than 2 times as large as the proportion of decreasing weeks. The proportion of decreasing weeks is low compared with that in the observations. Specifically, decreasing weeks occupy only 23.0–31.0% of the total forecasts by ECMWF. NCEP shows similar results, except for forecasting at a lead time of 5 weeks. This underestimation of the proportion of decreasing weeks is more severe in CMA. Moreover, the most severe underestimations of the proportion of decreasing weeks are the forecasts with a lead time of 2 or 3 weeks for all models.

The above results indicate that the models underestimate the frequency of TPSC dissipation, whereas they overestimate the frequency of TPSC accumulation, which leads to a systematic TPSC bias. To highlight increases in the overall TPSC biases, as well as changes in biases in successive weeks, a composite analysis is performed for all TPSC reforecasts during winter (Fig. 5a), increasing TPSC cases (Fig. 5b) and decreasing TPSC cases (Fig. 5c). All reforecasts initialized in winter are taken into account for the composite of all cases shown in Fig. 5a. The sample numbers of all cases are 187. Among all cases, we further select the increasing TPSC cases and decreasing TPSC cases. If the TPSC index continues to increase (decrease) for three weeks, this case is regarded as an increasing (decreasing) TPSC case. There are 46 increasing TPSC cases and 53 decreasing TPSC cases. We average the 46 (53) cases for different lead times. To focus on the increase in biases, values with a lead time of 1 week are removed for forecasting at all lead times.

On a seasonal average, the growth of the TPSC index in winter is only 1.3% over two weeks in the observation (black line in Fig. 5a). However, the models tend to exaggerate the growth of the TPSC index (colour lines in Fig. 5a). The growth of the TPSC index over the two weeks in the models ranged from 4.9% (ECMWF) to 9.8% (CMA). The TPSC index in the forecast shows distinct differences between the increasing TPSC cases and decreasing TPSC cases (Fig. 5b and 5c). The growth

of the TPSC index in the increasing TPSC cases is 14.1% over two weeks in the observation (black line in Fig. 5b). The growth of the TPSC index over two weeks in NCEP and CMA is close to that in the observation, while there is some underestimation of such growth in the ECMWF (colour lines in Fig. 5b). Although there are some differences between the TPSC index in the models and that in the observation, all models can forecast the increasing trend in the TPSC index. However, the situation for the decreasing TPSC cases is quite different. The reduction of the TPSC index in the decreasing TPSC cases is −10.0% over

two weeks in the observation (black line in Fig. 5c). However, all the changes in the TPSC index in the models are positive values (colour lines in Fig. 5c), indicating that there are some difficulties for the models in forecasting the dissipation of TPSC.

Studies have shown that current state-of-the-art atmospheric general circulation models (GCMs) tend to strongly overestimate the precipitation over the Tibetan Plateau (e.g., Su et al., 2013; Chen and Frauenfeld, 2014; Zhang et al., 2016; Zhang et al., 2019). For example, Su et al. (2013) evaluated 24 GCMs that were available in the fifth phase of the Coupled

Model Intercomparison Project (CMIP5) over the eastern Tibetan Plateau by comparing the model outputs with ground observations, and they found that all of the models consistently overestimated the observed precipitation for all seasons. Zhang et al., (2019) found similar results, in that all climate models they evaluated exaggerated the daily precipitation in the Tibetan Plateau during winter compared with the observed values. Here, we also found that the S2S models tended to overestimate the precipitation over the Tibetan Plateau. We compared the precipitation in the S2S models with both the gauges-based GPCC

precipitation dataset and the satellite-based TRMM precipitation dataset (Fig. 6). The regional averaging wintertime mean precipitation over the Tibetan Plateau in the GPCC and TRMM models are 0.27 mm day$^{-1}$ and 0.32 mm day$^{-1}$, respectively. Compared with the overserved precipitation, all three S2S models exaggerate the regional precipitation obviously. Notably, such an overestimation persists throughout the model integration. The ECMWF model reproduces the precipitation that is closest to the observations among the three models, but it still shows a large overestimation. The precipitation in the ECMWF

model is 0.78 mm day$^{-1}$ to 0.88 mm day$^{-1}$. The precipitation values in the NCEP model (1.07 mm day$^{-1}$ to 1.37 mm day$^{-1}$) and in the CMA model (1.50 mm day$^{-1}$ to 2.13 mm day$^{-1}$) have larger precipitation biases and even increase with the forecasting lead time. These overestimations of the precipitation induce underestimations of the TPSC dissipation, and they lead to positive biases in the TPSC from the models. Because the overestimation of the precipitation exists throughout the model integration, the positive biases of the TPSC accumulate and increase with the model integration.

In this section, it was found that S2S models underestimate the frequency of TPSC dissipation and have some difficulties forecasting TPSC dissipation with an observed rate. Exaggerations of the precipitation were found in all three models, which directly lead to accumulated overestimation of TPSC. As a result, systematic bias of TPSC occurs and increases with the model integration time.

# 4 Sensitivity of local surface air temperature to snow cover biases

## 4.1 Colder temperature with increasing forecast lead time

The local SAT over the Tibetan Plateau is highly correlated with simultaneous TPSC at a subseasonal time scale (Li et al., 2020a). Local snow-temperature relationships in S2S models were examined. We took a similar approach as in F. Li et al. (2019) and Diro and Lin (2020). The temporal correlation between the snow cover fraction and SAT with a lead of 1 week and 4 weeks for each grid point in the three models was computed to identify the extent and nature of the relationship (Fig. 7). Almost all of the regions exhibit a significant negative correlation in all of these three models. Additionally, such a relationship in all three models did not weaken with the forecasting lead time (compare Fig. 7a–c and Fig. 7d–f), even if the forecasting skill on the TPSC declined over time. The reason is that the relationship between the snow cover fraction and the SAT is embedded in the land surface model.

The skill of predicting the TPSC will further influence the skill of predicting the SAT. As shown in Section 3, the TPSC in the S2S models during the cold season increases with increasing forecast lead time. Such systematic biases of TPSC may influence the forecasted SAT in the S2S models. To test this hypothesis, we performed an analysis on SAT over the Tibetan Plateau similar to our analysis on TPSC. The SAT over the Tibetan Plateau is derived by averaging the SAT over the Tibetan Plateau region as defined in Section 2.2. Differences in the multiyear winter mean SAT over the Tibetan Plateau between forecasts with leads of 2–5 weeks and forecasts with leads of 1 week in the three models, which were obtained by subtracting the multiyear winter mean with a lead time of 1 week from that for forecast lead times of 2–5 weeks, are examined (Fig. 8). The differences in the three models show some common features. The differences in all three models are all negative. By comparing values at different lead times, we also find that such negative differences increase with increasing lead time, except for value at lead of 3 week in CMA model. The negative differences of SAT with the longest forecast lead time (5 weeks) are largest among all forecasts. The differences in SAT between the forecast for lead 5 weeks and the forecast for lead 1 week can be up to 1.9 °C. The increases in the SAT with the forecasting lead time are on a regional spatial scale (Fig. 9). Almost all of the grid points show negative values. Such increases in the CMA are less than those in ECMWF and NCEP.

The above results indicate that the SAT over the Tibetan Plateau becomes colder with increasing forecast lead time in the S2S models. Considering the results we obtained in Section 3, it can be concluded that the increasing TPSC is accompanied by decreasing SAT with forecast lead time.

## 4.2 Sensitivity of SAT to snow cover accumulation and dissipation

Section 3.2 reveals that models show different performances on snow cover accumulation and dissipation. We also found that there are some difficulties for the models in forecasting the dissipation of TPSC. To learn whether such different performances influence the SAT forecast and to examine the sensitivity of SAT to TPSC in the S2S models, we investigated the changes in SAT in the S2S models over the Tibetan Plateau during winter (Fig. 10a), as well as the increasing TPSC cases (Fig. 10b) and decreasing TPSC cases (Fig. 10c). To provide a SAT reference in the models, a composite was performed on SAT in the ERA-

interim reanalysis. We performed the same composite method as that is used in Section 3.2 on TPSC but for SAT over the Tibetan Plateau.

On a seasonal average, the change in SAT over the Tibetan Plateau in the reanalysis during winter is less than 0.1 °C (black line in Fig. 10a). However, the SAT in the models tends to decrease as the forecast lead time increases, especially in the ECMWF and NCEP models (colour lines Fig. 10a). The decline of the SAT over two weeks is 1.2 °C for the ECMWF and NCEP models. Considering the exaggerated growth of TPSC shown in Fig. 5a, a decrease in SAT is expected. In the ECMWF and NCEP models, more TPSC leads to lower SAT. SAT tends to be sensitive to TPSC in the ECMWF and NCEP models. However, SAT in the CMA model lacks sensitivity to TPSC. Although the exaggerated growth of the TPSC index in the CMA model is the most intense in these three models, the decrease in SAT in the CMA model is the least obvious.

The change in SAT should be closely connected to the variations in TPSC. The change in SAT in the increasing TPSC cases is −1.9 °C in two weeks in the ERA-interim reanalysis (black line in Fig. 10b), which is associated with the accumulation of TPSC (black line in Fig. 5b). SAT shows considerable decreases during the increasing TPSC cases (Fig. 10b). Cold biases of SAT between the forecasted SAT with lead time and that at the initial week tend to appear in all models (Fig. 10b), which is associated with accumulation of TPSC (in Fig. 5b). Here, the change in SAT in CMA over two weeks is smaller than that in the ECMWF and NCEP models. SATs in the ECMWF and NCEP models are more sensitive to TPSC than that in the CMA model.

Here, we further find that such biases lead to biases in SAT. SAT increases by 1.4 °C over two weeks in the reanalysis (black line in Fig. 10c), which is associated with the dissipation of TPSC (black line in Fig. 5c). However, the SATs in all these models shows small changes (colour lines in Fig. 10c) compared with that in the reanalysis. Such small changes in the SATs in the ECMWF and NCEP models are consistent with the changes in the TPSC indexes in these models, which show little changes (Fig. 5c). However, the large change in TPSC in the CMA model (Fig. 5c) does not induce large biases in SAT, indicating that the SAT in CMA lacks sensitivity to TPSC.

## 4.3 Numerical experiment

Through the results in Sections 4.1 and 4.2, we find that the local SAT over the Tibetan Plateau becomes colder with increasing forecast lead time. We assumed that the cold SAT biases are induced by the overestimation of TPSC. However, the relationship between snow cover and the atmosphere is a two-way coupling connection (Henderson et al., 2018). The assumption should be tested by numerical experiments (see Section 2.2 for details about the numerical model and experimental design). Otherwise, one may suspect that the cold SAT induces an increasing TPSC other than the TPSC influence on SAT. Therefore, we used the predicted TPSC as boundary condition in CTL runs (with overestimated TPSC), while observational TPSC in GDAS was used as boundary condition in EXP runs (without overestimated TPSC). The difference between CTL and EXP is considered to represent the response or the sensitivity of the SAT to the overestimated TPSC.

We averaged snow cover and SAT over the Tibetan Plateau in all simulations for CTL and EXP to obtain a composite for all reforecasts of TPSC during winter in the numerical experiment (Fig. 11a–b). As we discussed in Section 3.2, the growth

of the TPSC index in winter is only 1.3% for two weeks in the observations, while the S2S models tend to exaggerate the growth of the TPSC index (Fig. 5a). In the numerical experiment, CTL also exaggerates the growth of the TPSC index (blue line in Fig. 11a). Because both the NCEP S2S model and our numerical experiment use Noah as the land surface model, such biases may be attributed to the land surface model. Compared with CTL, EXP shows smaller cumulative biases (red line in Fig. 11a), which is because TPSC in EXP is replaced by TPSC in the FNL analyses every 6 hours. The SAT becomes colder with increasing forecast lead time in CTL (blue line in Fig. 11b). However, such a decrease in SAT is much smaller in EXP (red line in Fig. 11b). By checking the land surface energy fluxes over the Tibetan Plateau between CTL and EXP (Fig. 11c), we found that the overestimated TPSC strongly increases the upward-reflected shortwave radiation (negative value indicates enhanced upward radiation) due to the snow-albedo affect. This difference in the solar surface energy leads to a decrease in the absorbed solar radiation. Thus, the net shortwave radiation is decreased ($-10.2$ W m$^{-2}$), while the response of the net longwave radiation is much smaller than that of the net shortwave radiation. The decreased absorbed solar radiation is mainly balanced by the sensible heat flux (8.1 W m$^{-2}$; positive values indicates reduced upward heat flux). In contrast, the differences in the latent heat flux and ground heat flux are low. The overall responses of the surface energy to the overestimated TPSC lead to an incorrect cooling shift. Hence, the numerical experiment indicates that the cold SAT biases are induced by the overestimation of TPSC.

## 5 Conclusions and discussion

Accurate subseasonal-to-seasonal (S2S) atmospheric forecasts and hydrological forecasts have considerable socioeconomic value. This study evaluates the Tibetan Plateau snow cover (TPSC) prediction capabilities of three S2S forecast models (ECMWF, NCEP and CMA) during wintertime. These three S2S models can skilfully forecast TPSC variations within a lead time of 2 weeks during wintertime with temporal correlation coefficients greater than 0.5. ECMWF better captures TPSC variations compared with NCEP and CMA at a lead time of 1–2 weeks. All models show limited skill in forecasting TPSC at a lead time of 3 weeks or more. Compared with the IMS snow cover analysis, all three models tend to overestimate the area of TPSC. Another remarkable issue regarding the TPSC forecast is the increasing TPSC with forecast lead time, which makes the systematic positive biases of TPSC in models further increase at longer forecast lead times.

S2S models underestimate the frequency of TPSC dissipation, whereas they overestimate the frequency of TPSC accumulation. The accumulation and dissipation of wintertime TPSC occurs evenly in the observations. However, this kind of balance does not exist in the S2S models. In the models, the proportion of TPSC accumulation is mostly more than 2 times as large as the dissipation proportion. The most severe underestimations of the dissipation proportions are the forecasts at a lead time of 2 or 3 weeks for all models. The models also have some difficulties forecasting the TPSC dissipation at an observed rate. The growth of TPSC in the decreasing TPSC cases is $-10.0\%$ over two weeks in the observations, but all the changes in TPSC in the models are increasing.

All of the three S2S models consistently exaggerate the precipitation over the Tibetan Plateau compared to the observations. The exaggeration of the precipitation is prominent and always exists throughout the model integration. Systematic bias in the TPSC therefore occurs and accumulates with the model integration time due to exaggeration of the precipitation in the models.

    The increasing TPSC is accompanied by decreasing surface air temperature (SAT) with forecast lead time. The SAT over
the Tibetan Plateau becomes colder with increasing forecast lead time in the S2S models. The differences in SATs between the forecast for a lead of 5 weeks and the forecast for a lead of 1 week can be up to 1.9 °C. SATs tends to be sensitive to the TPSCs in both ECMWF and NCEP. However, SAT in CMA lacks sensitivity to TPSC. Numerical experiments were performed to test whether the cold SAT biases are induced by the TPSC overestimation. The control run exaggerates the growth of TPSC, which is similar to that in S2S models. The SAT in the control run becomes colder with integration time. When the increasing
TPSC with forecast lead time in the models along with the integration of the model is removed in the sensitivity run, the decreasing SAT with integration time also disappears. The overall responses of the surface energy to the overestimated TPSC lead to incorrect cooling shifts. This finding indicates that cold SAT biases are induced by the TPSC overestimation.

    Land–atmosphere coupling is one of the key physical processes for S2S prediction but is not well simulated and may reduce S2S prediction skill (Robertson et al., 2014; Dirmeyer et al., 2019). Studies have shown that better snow cover
initialization improves subseasonal and seasonal forecasts/simulations (Jeong et al., 2013; Orsolini et al., 2013; Senan et al., 2016; Lin et al., 2016; Kolstad, 2017; F. Li et al., 2019). This study indicates that in addition to snow cover initialization, a better model skill for snow cover prediction may also improves S2S prediction skill. More work is necessary and valuable to improve the prediction ability of models for snow cover.

    Previous studies have shown that current state-of-the-art GCMs tend to strongly overestimate the precipitation over the
Tibetan Plateau (e.g., Su et al., 2013; Chen and Frauenfeld, 2014; Zhang et al., 2016; Zhang et al., 2019). It is worthwhile to note that the S2S models also significantly overestimate the precipitation over the Tibetan Plateau and further cause other biases (e.g., TPSC biases and SAT biases). It is of great significance to reduce the biases of the precipitation over the Tibetan Plateau in the GCMs. Surface winds and snow sublimation could also play a role in causing the snow ablation. Identifying the relative contributions of these factors to the biased snow prediction needs more detailed and careful diagnoses. Note that the
current study analysed the data during common reforecast period of 1999–2010 for ECMWF, NCEP and CMA models. All these three operational models provide real time forecast since 2015 based on the improved prediction systems. It could be valuable to carry out evaluation works based on the up-to-date forecast results. Future studies on these issues are potentially valuable.

**Data and code availability**

The data and model used in this study are free to the public. The S2S datasets and ERA-interim data are available at https://apps.ecmwf.int/datasets/. The IMS snow cover data are available at https://nsidc.org/data/G02156. The GPCC data are

available at https://www.dwd.de/EN/ourservices/gpcc/gpcc.html. The TRMM data are available at https://disc.gsfc.nasa.gov. The NCEP FNL data are available at https://rda.ucar.edu/datasets/ds083.2/. The WRF source codes can be obtained at https://www2.mmm.ucar.edu/wrf/users/download/get_source.html. All figures were produced using NCAR Command Language (NCL) version 6.6.2, an open source software free to the public, by UCAR/NCAR/CISL/TDD, https://doi.org/10.5065/d6wd3xh5. The NCL scripts used in this study are available from the corresponding author upon reasonable request.

**Author contribution**

W.L. led the overall scientific questions and designed the research. S.H. and W.L. analyzed the data and drafted the manuscript for initial submission. W.L. analyzed the data for the revised manuscript. W.L., P.H., W.G. and J.W. made substantial contributions to revise the manuscript and prepare the responses to the referees.

**Competing interests**

The authors declare no competing interests.

**Acknowledgements**

This research is supported by the National Key Research and Development Program of China (2018YFC1505804), Natural Science Foundation of China (41905074), the Natural Science Foundation of Jiangsu Province (BK20190782).

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

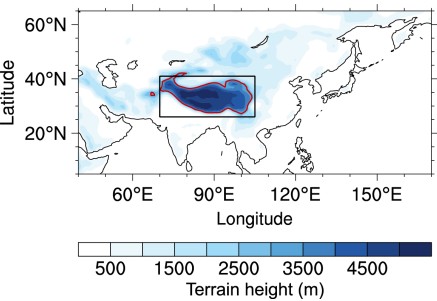

**Figure 1.** The location and topography of the Tibetan Plateau. Shading shows topography (unit: m). The black rectangle shows the region within 26–41°N and 70–105°E. The red contour marks altitudes at 3000 m. The Tibetan Plateau area, which is the focus of this study is the region within the black rectangle at an altitude of greater than 3000 m. This figure also shows the simulation domain for numerical experiments in this study. The map in this figure was generated using NCAR Command Language (NCL) version 6.6.2, an open source software free to the public (doi: 10.5065/D6WD3XH5).

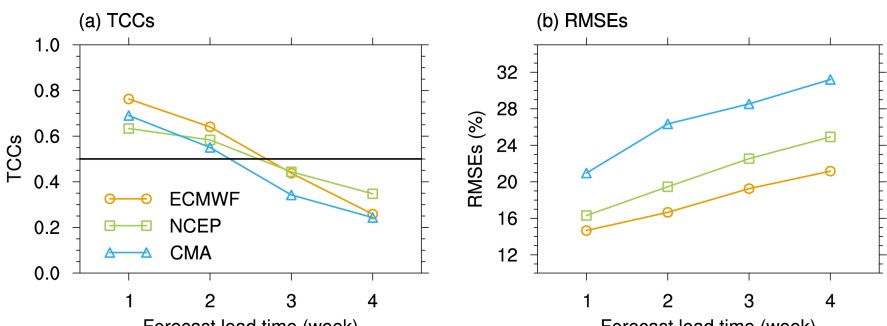

**Figure 2.** Prediction skill of the Tibetan Plateau snow cover (TPSC) index in the S2S models during wintertime. **(a)** The temporal correlation coefficients (TCCs; *y*-axis) between the observed TPSC index and the predicted TPSC index in the ECMWF (orange line), NCEP (green line) and CMA (blue line) models during winter. The *x*-axis represents the forecast lead time (unit: week). A good prediction skill has a TCC that is greater than 0.5 (marked by black line). **(b)** is similar to **(a)** but is for the root-mean-square errors (RMSEs; *y*-axis, unit: %).

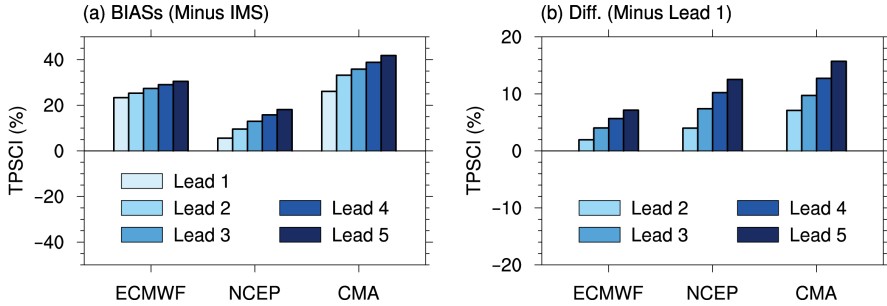

**Figure 3. (a)** The multiyear wintertime mean biases of Tibetan Plateau snow cover (TPSC) index (unit: %) in forecasts against those in the Interactive Multisensor Snow and Ice Mapping System (IMS) snow cover analysis. **(b)** Differences in the multiyear wintertime mean TPSC index between forecasts with a lead of 2–5 weeks and forecasts with a lead of 1 week in each model.

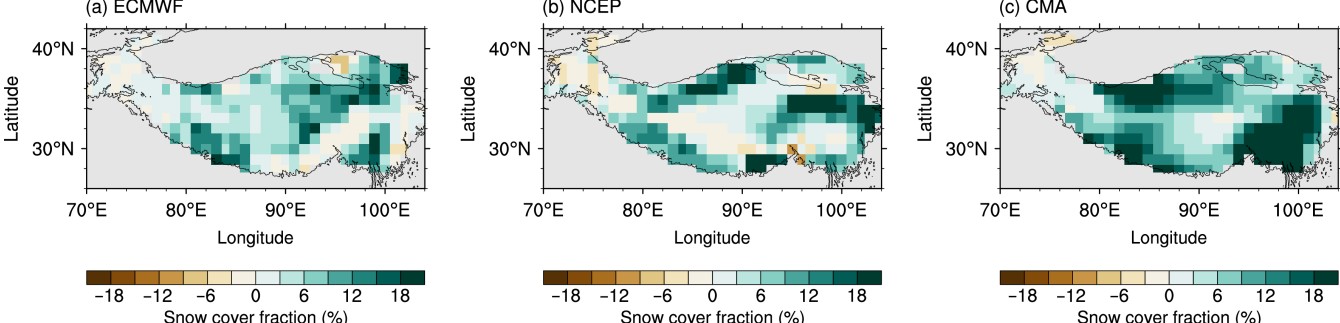

**Figure 4.** Differences in the multiyear wintertime mean Tibetan Plateau snow cover fraction (unit: %) between forecasts with a lead of 4
605 weeks and forecasts with a lead of 1 week in **(a)** ECMWF, **(b)** NCEP and **(c)** CMA. The map of Tibetan Plateau was created by using Global Relief Model data of ETOPO1, topographic data free to the public (doi:10.7289/V5C8276M).

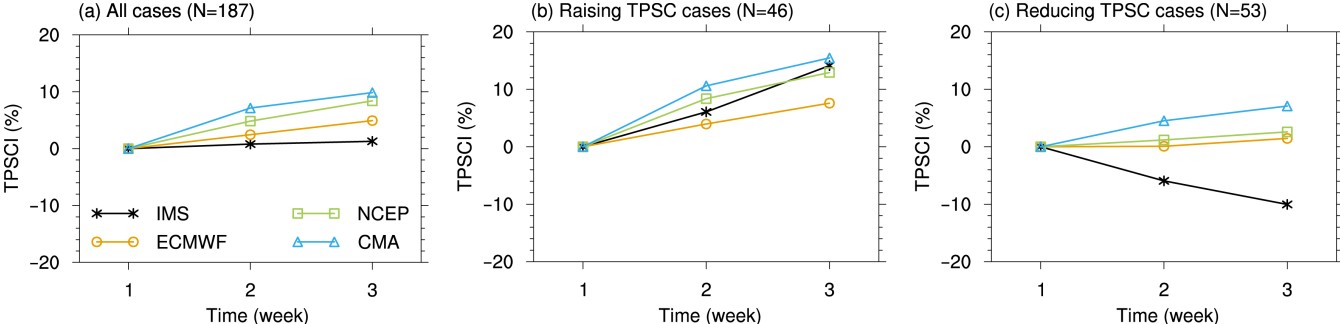

**Figure 5.** Composites of the Tibetan Plateau snow cover index (unit: %) for **(a)** all cases, **(b)** increasing TPSC cases, and **(c)** decreasing TPSC cases. Black lines, orange lines, green lines and blue lines represent composites in the observation, ECMWF, NCEP, and CMA, respectively; see legend in (a). The *x*-axis represents the number of weeks in the cases for the composites, which are also forecast lead times (unit: week). The "N" in the title of each plot indicates the number of cases for the composite.

**Figure 6.** The multiyear wintertime mean precipitation over the Tibetan Plateau (unit: mm day$^{-1}$) for the observations and forecasts in each model.

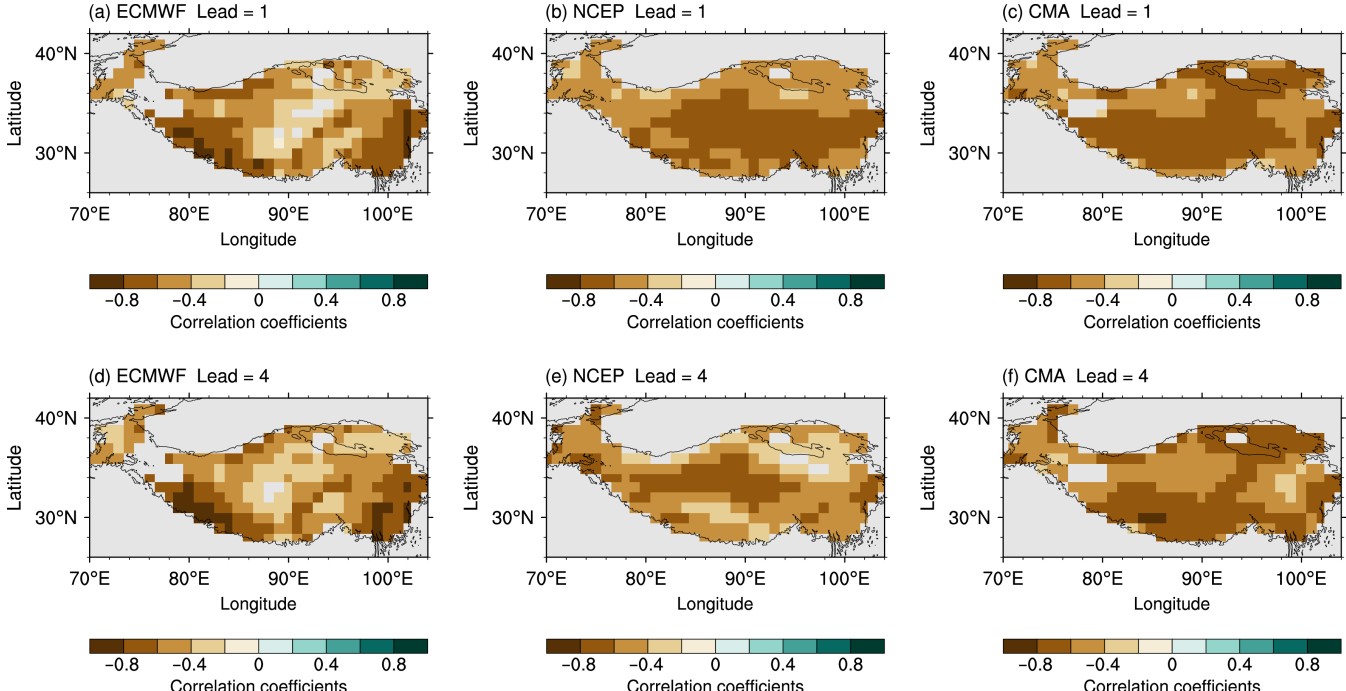

**Figure 7.** Spatial pattern of correlations between the snow cover fraction and the surface air temperature with a lead of 1 week in **(a)** ECMWF, **(b)** NCEP and **(c)** CMA. Only significant correlations at the 0.01 level are displayed. (d)–(f) is similar to (a)–(c) but for forecasting with a lead of 4 weeks. The map of Tibetan Plateau was created by using Global Relief Model data of ETOPO1, topographic data free to the public (doi:10.7289/V5C8276M).

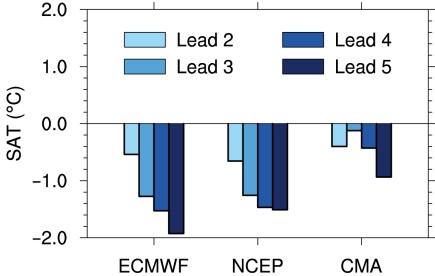

**Figure 8.** Differences in the multiyear wintertime mean surface air temperature over the Tibetan Plateau (unit: °C) between forecasts with a lead of 2–5 weeks and forecasts with a lead of 1 week in each model.

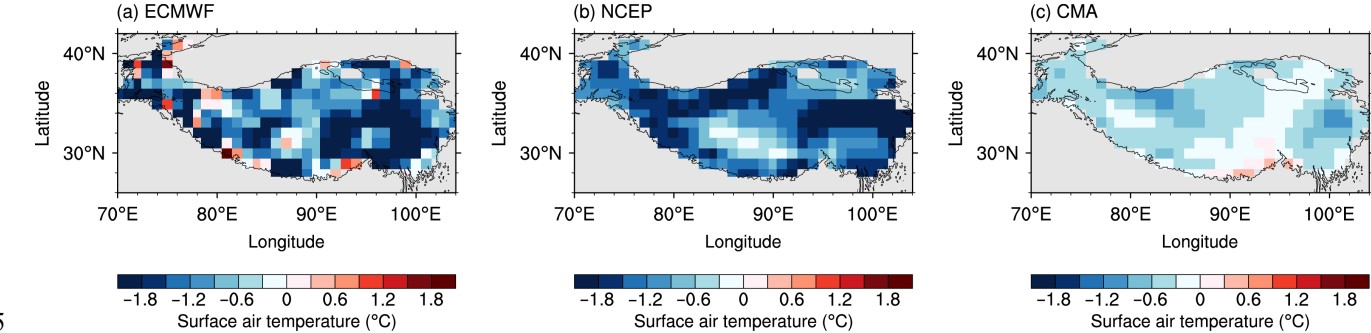

**Figure 9.** Differences in the multiyear wintertime mean surface air temperature over the Tibetan Plateau (unit: °C) between forecasts with a lead of 4 weeks and forecasts with a lead of 1 week in **(a)** ECMWF, **(b)**NCEP and **(c)** CMA. The map of Tibetan Plateau was created by using Global Relief Model data of ETOPO1, topographic data free to the public (doi:10.7289/V5C8276M).

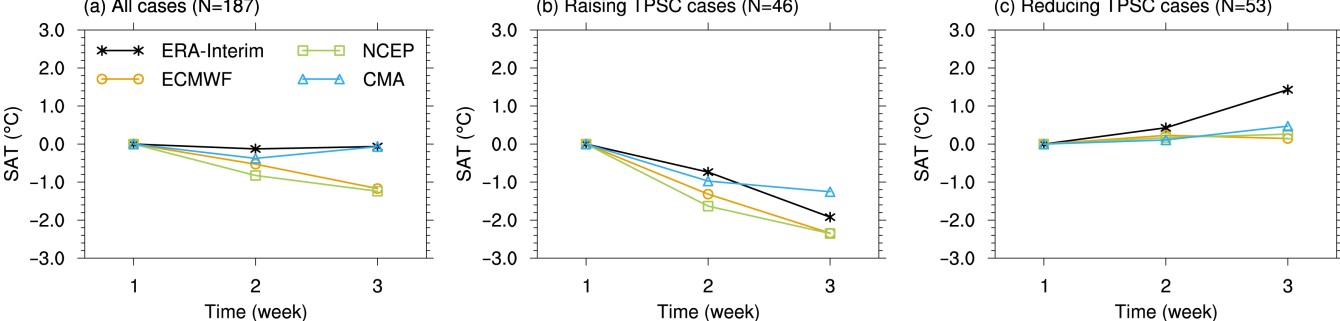

**Figure 10.** Composites of surface air temperature over the Tibetan Plateau (unit: °C) for (a) all cases, (b) increasing TPSC cases, and (c) decreasing TPSC cases. Black lines, orange lines, green lines, blue lines and black lines represent composites in observation, ECMWF, NCEP, and CMA, respectively; see legend in (a). The *x*-axis represents the number of weeks in the cases for the composites, which are also forecast lead times (unit: week). The "N" in the title of each plot indicates the number of cases for the composite.

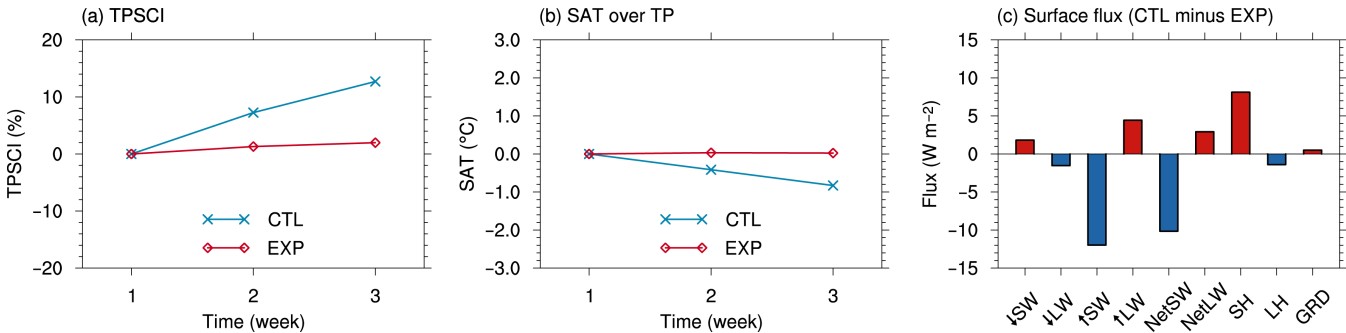

**Figure 11.** Sensitivity of SAT and surface energy balance to TPSC biases in the numerical experiments. (a) TPSCI and (b) SAT over the Tibetan Plateau in CTL (blue lines) and EXP (red lines) runs. The units of TPSCI and SAT are % and °C. The *x*-axis represents the number of weeks lagging the start of the model initial date (unit: week). (c) The difference in the surface energy balance between the CTL and EXP (CTL minus EXP) at 3 weeks in the numerical experiments. The terms from left to right are downward shortwave radiation (↓SW), downward longwave radiation (↓LW), upward shortwave radiation (↑SW), upward longwave radiation (↑LW), net shortwave radiation (NetSW), net longwave radiation (NetLW), sensible heat flux (SH), latent heat flux (LH) and ground heat flux (GRD) at the surface over the Tibetan Plateau, respectively (unit: W m$^{-2}$). The flux sign is positive downwards.

**Table 1.** The proportion of increasing (decreasing) weeks in the observations and forecast models with different lead times (in weeks).

|  | Observation/Lead=2 | Lead=3 | Lead=4 | Lead=5 |
|---|---|---|---|---|
| IMS | 50.3% (49.7%) | | | |
| ECMWF | 72.7% (27.3%) | 77.0% (23.0%) | 70.6% (29.4%) | 69.0% (31.0%) |
| NCEP | 77.5% (22.5%) | 69.5% (30.5%) | 64.7% (35.3%) | 55.1% (44.9%) |
| CMA | 86.6% (13.4%) | 67.4% (32.6%) | 72.2% (27.8%) | 79.7% (20.3%) |