# Peer review of "Systematic bias of Tibetan Plateau snow cover in subseasonal-toseasonal models"

_The Cryosphere, 2020_

## Referee Comment (RC1) · Anonymous Referee #1 · 19 Jun 2020

Intraseasonal variation of snow cover over Tibetan Plateau is very important for the prediction of surrounding and downstream regions. Recognizing the subseasonal prediction skill of TP snow cover in the current models are crucial for correcting and improving subseasonal prediction. Snow cover's S2S skill is scarcely studied, which is worthwhile to investigate. However, the current version has large space to improve. I suggest the resubmission after reframing the writing and clarifying the following points. 1. The writing frame should be modified. e.g., a. a method part should introduce the major method how to evaluate the S2S skill; b. the numerical experiment design and modeling introduction should be put earlier in this manuscript. 2. The evaluation method of S2S skill is conventional and simple. To me, the major contribution of this

study is intentionally S2S evaluation. therefore, please give some quantitative evaluation rather than only TCC. 3. What season of this study is focused on? I cannot find any information for this. Meanwhile, I guess the S2S prediction skill should have large seasonal dependence even monthly dependence. Please check. 4. Regional modeling portion, I cannot understand it very well. To me, one is the predicted lateral boundary layer, the other is observational boundary layer, of course the latter is better than the former. I don't know which point does this study want to present through the numerical modeling. 5. To fit "Cryosphere", which is high-quality journal, at least, some physical analysis are needed. e.g., land-air budget analysis (surface fluxes) should be added to interpret the linkage between snow cover and surface temperature.

---

## Referee Comment (RC2) · Anonymous Referee #2 · 21 Jul 2020

This paper examines how a set of state-of-the-art subseasonal-to-seasonal (S2S) forecast systems predict the Tibetan Plateau-wide snow cover and surface temperature over the 1999-2010 period. The forecast systems from ECMWF, CMA and NCEP are used in the intercomparison. The evaluation of forecasted snow cover is made against the multi-instrument (IMS) snow cover data. A connection is drawn between the bias is snow cover, which increases systematically with lead time in all models, and a bias in surface temperature. Model experiments with the WRF model support the idea that the latter is induced by the snow excess through land-atmosphere coupling. Few papers have examined snow forecast on the S2S time scale on the Tibetan Plateau (TP), a region with well-known biases in snow and surface temperature. This is an innovative

study that is well-worthy of publication in The Cryosphere. I however recommend a major revision of the paper, before it is in an acceptable form for publication.

MAJOR COMMENTS 1) Some brief description of how the three different models are initialised in terms of snow and land surface is needed, complementing the description of the land-surface models. What sort of snow analysis is used to initialise the different models? Isn't ERA-5, which has strong snow biases over the TP, used to initialise the ECMWF S2S reforecasts? Which observations are used in these snow analyses, both globally and over the TP more specifically? While IMS is not used to initialise the ECMWF forecast model over the TP (see Orsolini Y. et al., 2019), is it used in the other systems? The quality of this initialisation would most certainly influence the snow forecasts at the S2S time scale. The initial snow values may be a key addition to bring in Fig 3. 2) Some more details about how the snow cover fraction conversion is derived in each model is needed. For example, a 100% snow cover may mean very different snow depth or snow water equivalent (the actual prognostic variables) in the different prediction systems. Also, I believe that IMS provides a binary information snow cover being 0 or 1, with the former case meaning that the fractional snow cover is below 50%. When aggregated to a 1-degree grid (L88), isn't there a range or uncertainty in the IMS aggregated value (given that the observed fractional snow cover could actually be 0 or 50%)? Please clarify these points and the implications for forecast verification. 3) The authors consider the snow cover averaged over the entire Tibetan-Plateau, but they do not show any snow cover maps although there is considerable heterogeneity, as shown by the authors in previous publications. I wonder if there could be compensation effects between different geographical sub-regions, that could result in an agreement of the TPSC index between forecasts and IMS. Different prediction systems may have different regional biases. Showing such maps would re-assure the reader that the prediction systems capture the main climatological features of the snow distribution over the TP and its S2S variability (e.g., in subregions as in Li W. et al., IJOC, 2019). Could spatial pattern correlation between IMS and snow forecasts be helpful in this case? 4) The authors could also try to examine the snow-temperature coupling strength (as an
indicator of land-atmosphere interaction) in the respective prediction systems by corre-lating the local forecasted temperature and forecasted snow, as a function of lead time. See Diro and Lin (2020) or Li et al. (2019).

Diro, G. T., and H. Lin, 2020: Subseasonal Forecast Skill of Snow Water Equivalent and Its Link with Temperature in Selected SubX Models. Wea. Forecasting, 35, 273–284. Orsolini, Y., et al. G. (2019), Evaluation of snow depth and snow cover over the Tibetan Plateau in global reanalyses using in situ and satellite remote sensing ob-servations. The Cryosphere,Vol. 13.(8) s.2221-2239 Li, F., et al. (2019). Impact of snow initialization in subseasonal‐to‐seasonal winter forecasts with the Norwe-gian Climate Prediction Model. Journal of Geophysical Research: Atmospheres, 124. https://doi.org/10.1029/2019JD030903

MINOR COMMENTS L92-93: the description here focuses on winter. While winter is the main focus, many plots show year-long results. It would be better to emphasize the whole set used (nb of years, total nb of forecasts,. . .), not only the winters. L252: the snow bias might not only arise from the land surface model (shared between WRF and the NCEP model) but also from the meteorological forcing (e.g., excess precipitation). Please clarify. L244: how is the GDAS snow analysis used in Section 4.3 on numerical modelling compare with the IMS snow analysis, used in the first part of the paper. While it is mentioned that GDAS assimilates IMS, does it assimilate it over the TP? What does it assimilate specifically over the TP (in-situ data?)? Would the prediction skill be different if evaluated against GDAS (Fig 1) ? Conclusions: a brief mention of possible, relevant physical processes over the TP leading to snow ablation would be helpful. Could it be the strong surface winds or else the snow sublimation missing in the models? The short length of the period over which the forecasts are evaluated (around 10 years) is a bit of a concern. It appears that the biases are quite strong and systematic, but I wonder if some features in the forecasts would be robust over a longer period: for example, the slowdown in TPSC in early winter (December) seen in ECMWF and NCEP (Fig 3). I realise that if adding another 10 years may entail a lot of

computational work, but it would add to the robustness of the conclusions. At least, a word of caution in the summary is warranted.

Typos / English L43: hydrologic cycle L28: radiative rather than radiant L108: the total variability L113: and the three different L143: the preceding week rather than the last week, seems more appropriate L152: accumulation, leading to a systematic TPSC bias. L168: growth is used for a declining variable. Either decrement, reduction or decline should be clearer. L173: real rate should be observed rate or rate derived from IMS. L173: indices or index L202: growth of SAT, rather than TPSC (it says 1.2 degree). L229: land-atmosphere interactions L256: Hence,

---

## Author Comment (AC1) · 26 Aug 2020

Dear Reviewer,

We would like to thank you for your constructive and helpful comments, which helped us to improve our manuscript. Significant changes have been made according to your comments and suggestions.

The following is our point-by-point response. The reviewer's comments are shown in *blue italics*. Our responses are provided in **black**. The revised text is in **red**.

Sincerely,
All of the authors
* * *
*Reviewer #1*

*Intraseasonal variation of snow cover over Tibetan Plateau is very important for the prediction of surrounding and downstream regions. Recognizing the subseasonal prediction skill of TP snow cover in the current models are crucial for correcting and improving subseasonal prediction. Snow cover's S2S skill is scarcely studied, which is worthwhile to investigate. However, the current version has large space to improve. I suggest the resubmission after reframing the writing and clarifying the following points.*

*1. The writing frame should be modified. e.g., a. a method part should introduce the major method how to evaluate the S2S skill; b. the numerical experiment design and modeling introduction should be put earlier in this manuscript.*

**Response:**

Good suggestions. Your suggested writing frame looks much more logic and clear. In the revised manuscript, Section 2 is now "2 Data and method", which contains "2.1 S2S forecast models", "2.2 Validation data and method", "2.3 Numerical model and experimental design".

*2. The evaluation method of S2S skill is conventional and simple. To me, the major contribution of this study is intentionally S2S evaluation. therefore, please give some quantitative evaluation rather than only TCC.*

**Response:**

Following your suggestions, three evaluation metrics, including the temporal correlation coefficient (TCC), the root-mean-square error (RMSE), and the mean bias are used to quantify the subseasonal forecast skill of TPSC in the state-of-the-art S2S models in the revised manuscript. In addition to these simple metric assessments, a composite analysis for increasing and decreasing TPSC cases is performed to further understand what leads to the forecast biases, which is crucial for model developers and users. Spatial pattern of systematic biases of TPSC for each grid points have also been provided in the revised manuscript.

*3. What season of this study is focused on? I cannot find any information for this. Meanwhile, I guess the S2S prediction skill should have large seasonal dependence even monthly dependence. Please check.*

**Response:**

Many thanks for your comments. This issue was also raised by Reviewer #2. We actually intended to focus on TPSC assessment in boreal winter, but our presentation in the original manuscript might not be clear and cause some confusion. Unlike the systematic biases of wintertime TPSC revealed by the S2S models, the forecast errors in summer are not consistent and show complex structures among different models. Thus, we focus on only the winter season in the current study and will leave the issues about summertime TPSC prediction for our future work.

*4. Regional modeling portion, I cannot understand it very well. To me, one is the predicted lateral boundary layer, the other is observational boundary layer, of course the latter is better than the former. I don't know which point does this study want to present through the numerical modeling.*

**Response:**

The relationship between snow cover and the atmosphere is a two-way coupling connection. Model sensitivity experiment is a good tool to clarify cause and effect. By designing and conducting the model experiments, we attempted to verify whether the cold SAT biases predicted by S2S models were caused by the overestimation of TPSC (instead of the opposite condition that the cold SAT leads to overestimated TPSC). Therefore, we used the predicted TPSC as boundary condition in CTL runs (with overestimated TPSC), while observational TPSC in

GDAS was used as boundary condition in EXP runs (without overestimated TPSC). We clarified the purpose why we carried out the numerical experiments in the revised manuscript.

"To reveal the causality of the systematic bias of the TPSC-induced regional SAT bias, numerical experiments are performed." (in the revised Section 2)

"Through the results in Sections 4.1 and 4.2, we find that the local SAT over the Tibetan Plateau becomes colder with increasing forecast lead time. We assumed that the cold SAT biases are induced by the overestimation of TPSC. However, the relationship between snow cover and the atmosphere is a two-way coupling connection. The assumption should be tested by numerical experiments (see Section 2.2 for details about the numerical model and experimental design). Otherwise, one may suspect that the cold SAT induces an increasing TPSC other than the TPSC influence on SAT. Therefore, we used the predicted TPSC as boundary condition in CTL runs (with overestimated TPSC), while observational TPSC in GDAS was used as boundary condition in EXP runs (without overestimated TPSC). The difference between CTL and EXP is considered to represent the response or the sensitivity of the SAT to the overestimated TPSC." (in the revised Section 4)

*5. To fit "Cryosphere", which is high-quality journal, at least, some physical analysis are needed. e.g., land-air budget analysis (surface fluxes) should be added to interpret the linkage between snow cover and surface temperature.*

**Response:**

Excellent comments. We diagnosed the surface energy budget equation (Fig. 11c in the revised manuscript), the results of which indeed provides insightful explanations of TPSC-SAT relationship.

"By checking the land surface energy fluxes over the TP between CTL and EXP (Fig. 11c), we found that the overestimated TPSC strongly increases the upward-reflected shortwave radiation due to the snow-albedo affect. This difference in the solar surface energy leads to a decrease in the absorbed solar radiation. Thus, the net shortwave radiation is decreased ($-10.2$ W m$^{-2}$), while the response of the net longwave radiation is much smaller than that of the net shortwave radiation. The decreased absorbed solar radiation is mainly emitted by the land surface as sensible heat flux ($-8.1$ W m$^{-2}$). In contrast, the differences in the latent heat flux and ground heat flux are low. The overall responses of the surface energy to the overestimated TPSC lead to an incorrect cooling shift." (in the revised Section 4)

[Figure]

**Figure 11c in the revised manuscript**. Sensitivity of surface energy balance to TPSC biases in the numerical experiments. The difference in the surface energy balance between the CTL and EXP (CTL minus EXP) at 3 weeks in the numerical experiments. The terms from left to right are downward shortwave radiation ($\downarrow$SW), downward longwave radiation ($\downarrow$LW), upward shortwave radiation ($\uparrow$SW), upward longwave radiation ($\uparrow$LW), net shortwave radiation (NetSW), net longwave radiation (NetLW), sensible heat flux (SH), latent heat flux (LH) and ground heat flux (GRD) at the surface over the TP, respectively (unit: W m$^{-2}$).

---

## Author Comment (AC2) · 26 Aug 2020

Dear Reviewer,

We would like to thank you for your constructive and helpful comments, which helped us to improve our manuscript. Significant changes have been made according to your comments and suggestions.

The following is our point-by-point response. The reviewer's comments are shown in **_blue italics_**. Our responses are provided in **black**. The revised text is in **red**.

Sincerely,
All of the authors
* * *
*Reviewer #2*

*This paper examines how a set of state-of-the-art subseasonal-to-seasonal (S2S) fore- cast systems predict the Tibetan Plateau-wide snow cover and surface temperature over the 1999-2010 period. The forecast systems from ECMWF, CMA and NCEP are used in the intercomparison. The evaluation of forecasted snow cover is made against the multi-instrument (IMS) snow cover data. A connection is drawn between the bias is snow cover, which increases systematically with lead time in all models, and a bias in surface temperature. Model experiments with the WRF model support the idea that the latter is induced by the snow excess through land-atmosphere coupling. Few papers have examined snow forecast on the S2S time scale on the Tibetan Plateau (TP), a region with well-known biases in snow and surface temperature. This is an innovative study that is well-worthy of publication in The Cryosphere. I however recommend a major revision of the paper, before it is in an acceptable form for publication.*

*MAJOR COMMENTS*
*1) Some brief description of how the three different models are initialised in terms of snow and land surface is needed, complementing the description of the land-surface models. What sort of snow analysis is used to initialise the different models? Isn't ERA-5, which has strong snow biases over the TP, used to initialise the ECMWF S2S reforecasts? Which observations are used in these snow analyses, both globally and over the TP more specifically? While IMS is not used to*

*initialise the ECMWF forecast model over the TP (see Orsolini Y. et al., 2019), is it used in the other systems? The quality of this initialisation would most certainly influence the snow forecasts at the S2S time scale. The initial snow values may be a key addition to bring in Fig 3.*

**Response:**

Thank you for the useful comments. We completely agree with the reviewer that the information of snow initializations in S2S reforecasts has to be described clearly because they are the key factors influencing the TPSC prediction – the topic of our study. With a careful check, we confirmed that the northern hemisphere IMS snow cover data and ground observations of the snow depth available on the Global Telecommunications System (GTS) are used to initialize snow in the ECMWF S2S model. This is different from snow initialization in the ERA-5. Descriptions of snow initializations in the S2S forecasts were added in revised manuscript.

"For the ECMWF model, realistic snow is initialized in the forecasts. The snow mass is initialized by the ECMWF Land Data Assimilation System (LDAS). The snow initialization relies on a land surface synoptic report and national ground observations of the snow depth available on the Global Telecommunications System (GTS), as well as on the Interactive Multisensor Snow and Ice Mapping System (IMS) snow cover information. The NCEP model also initialize realistic snow in the forecasts. The snow initialization comes from the Climate Forecast System Reanalysis snow analysis using IMS and the Air Force Weather Agency snow depth analysis. Snow in the CMA model was not directly initialized in the forecasts. The initial conditions of the snow in the CMA model are from a balanced state produced by long-term air-sea initialization integration. See the details on snow initialization in the S2S models at https://confluence.ecmwf.int/display/S2S/Models." (in the revised Section 2)

*2) Some more details about how the snow cover fraction conversion is derived in each model is needed. For example, a 100% snow cover may mean very different snow depth or snow water equivalent (the actual prognostic variables) in the different prediction systems. Also, I believe that IMS provides a binary information snow cover being 0 or 1, with the former case meaning that the fractional snow cover is below 50%. When aggregated to a 1-degree grid (L88), isn't there a range or uncertainty in the IMS aggregated value (given that the observed fractional snow cover could actually be 0 or 50%)? Please clarify these points and the implications for forecast verification.*

**Response:**

1. We added detailed descriptions about how the snow cover fraction conversion is derived in each model in the revised Section 2.1.

"According to the snow scheme in each land surface model, we obtain the snow cover fraction, which is a diagnostic variable in this study.

The snow cover fraction ($f_{snow}$) in the ECMWF model is parameterized as follows:

$$f_{snow} = min[\ 1,\ S/(0.1 \times \rho)\ ] \tag{1}$$

where $min$ indicates the minimum function, $S$ is the snow water equivalent (unit is kg m$^{-2}$), and $\rho$ is the snow density (unit is kg m$^{-3}$) (Dutra et al., 2010).

The $f_{snow}$ in the NCEP model is parameterized as follows:

$$f_{snow} = min[\ 1,\ 1 - (e^{-0.001 \times 2.6 \times S/SNUP} - 0.001 \times S/SNUP \times e^{-2.6})\ ] \tag{2}$$

where $e$ is the natural logarithm, and $SNUP$ is the vegetation parameter, which indicates the threshold snow depth (in water equivalent m) that implies 100% snow cover (Koren et al., 1999; Ek et al., 2003). The $SNUP$ ranges from 0.01 to 0.08 for different vegetation types. Details on the Noah code and vegetation parameters can be accessed in https://ral.ucar.edu/solutions/products/unified-noah-lsm.

The $f_{snow}$ in the CMA model is parameterized as follows:

$$f_{snow} = min[\ 1,\ 1.77 \times d/(d + 10.6)\ ] \tag{3}$$

where $d$ is the snow depth (unit is cm), which is calculated from the snow water equivalent and snow density (Wu and Wu, 2004)."

2. We used a method similar to Orsolini (2019) to aggregate a 24-km resolution IMS analysis, which is interpolated into the 1°×1° grid of the S2S models. Details are provided in revised Section 2.2.

"The original 24-km resolution IMS analysis is interpolated into the 1°×1° grid of the S2S models. IMS provides binary snow cover information: it has the value of 1 if more than 50% of the 24-km pixel is covered by snow; otherwise, it is 0 (snow free). Orsolini et al. (2019) aggregated the original IMS product to a lower resolution rectilinear grid. They counted the number of pixels with a value of 1 in a grid box; assuming that they have 100% cover gave the high estimate, and assuming that they represent 50% cover gave the low estimate. These two estimates provide a range of values, which reflects the uncertainty inherent to aggregating the 24-km binary data, e.g., a value of 1 in a pixel means a 50% to 100% snow coverage. Here, we

used a method similar to Orsolini et al. (2019) to interpolate the original IMS product into the 1°×1° grid of S2S products, but we further averaged these two estimates."

*3) The authors consider the snow cover averaged over the entire Tibetan-Plateau, but they do not show any snow cover maps although there is considerable heterogeneity, as shown by the authors in previous publications. I wonder if there could be compensation effects between different geographical sub-regions, that could result in an agreement of the TPSC index between forecasts and IMS. Different prediction systems may have different regional biases. Showing such maps would re-assure the reader that the prediction systems capture the main climatological features of the snow distribution over the TP and its S2S variability (e.g., in subregions as in Li W. et al., IJOC, 2019). Could spatial pattern correlation between IMS and snow forecasts be helpful in this case?*

**Response:**

Good suggestions. We plotted additional figures in the revised manuscript to show the spatial pattern of the systemic bias (Fig. 4 in the revised manuscript), as suggested by the reviewer.

"Taking the differences between the forecasts with a lead of 4 weeks and the forecasts with a lead of 1 week as an example, the spatial patterns of these increases in the biases in the three models show some similarities (Fig. 4). Although the spatial patterns of the differences in the three models show some small discrepancies, the differences are mainly positive in the three models, especially over parts of central and eastern Tibetan Plateau. These indicate that the increasing TPSC with the forecasting lead time are at a regional spatial scale." (in the revised Section 3)

[Figure]

**Figure 4 in the revised manuscript.** Differences in the multiyear wintertime mean Tibetan Plateau snow cover fraction (unit: %) between forecasts with a lead of 4 weeks and forecasts with a lead of 1 week in (a) ECMWF, (b) NCEP and (c) CMA.

*4) The authors could also try to examine the snow-temperature coupling strength (as an C2 indicator of land-atmosphere interaction) in the respective prediction systems by correlating the local forecasted temperature and forecasted snow, as a function of lead time. See Diro and Lin (2020) or Li et al. (2019).*

*Diro, G. T., and H. Lin, 2020: Subseasonal Forecast Skill of Snow Water Equivalent and Its Link with Temperature in Selected SubX Models. Wea. Forecasting, 35, 273– 284. Orsolini, Y., et al. G. (2019), Evaluation of snow depth and snow cover over the Tibetan Plateau in global reanalyses using in situ and satellite remote sensing ob- servations. The Cryosphere,Vol. 13.(8) s.2221-2239*

*Li, F., et al. (2019). Impact of snowinitializationinsubseasonalâ˘Rˇtoâ˘Rˇseasonalwinterforecastswiththe Norwe- gian Climate Prediction Model. Journal of Geophysical Research: Atmospheres, 124. https://doi.org/10.1029/2019JD030903*

**Response:**

Many thanks for your good suggestion. The temporal correlations between snow cover fraction and SAT with forecast lead times of 1 week and 4 week for each grid point in three models were computed to identify the extent and nature of the snow-temperature relationship (Fig. 7 in the revised manuscript).

"The local SAT over the Tibetan Plateau is highly correlated with simultaneous TPSC at a subseasonal time scale (Li et al., 2020a). Local snow-temperature relationships in S2S models were examined. We took a similar approach as in F. Li et al. (2019) and Diro and Lin (2020). The temporal correlation between the snow cover fraction and SAT with a lead of 1 week and 4 weeks for each grid point in the three models was computed to identify the extent and nature of the relationship (Fig. 7). Almost all of the regions exhibit a significant negative correlation in all of these three models. Additionally, such a relationship in all three models did not weaken with the forecasting lead time (compare Fig. 7a–c and Fig. 7d–f), even if the forecasting skill on the TPSC declined over time. The reason is that the relationship between the snow cover fraction and the SAT is embedded in the land surface model." (in the revised Section 4)

[Figure]

**Figure 7 in the revised manuscript.** Spatial pattern of correlations between the snow cover fraction and the surface air temperature with a lead of 1 week in (a) ECMWF, (b) NCEP and (c) CMA. Only significant correlations at the 0.01 level are displayed. (d)–(f) is similar to (a)–(c) but for forecasting with a lead of 4 weeks.

*MINOR COMMENTS*

*L92-93: the description here focuses on winter. While winter is the main focus, many plots show year-long results. It would be better to emphasize the whole set used (nb of years, total nb of forecasts,. . .), not only the winters.*

**Response:**

We are very sorry for the confusion. This issue is also raised by Reviewer #1. In fact, we attempted to focus on the TPSC study only for boreal winter considering that the TPSC biases during summer are not consistent among different models and there might have some complex processes involved. We have clarified the season that our present study focuses on and will leave the investigation of summertime TPSC prediction for our future work.

*L252: the snow bias might not only arise from the land surface model (shared between WRF and the NCEP model) but also from the meteorological forcing (e.g., excess precipitation). Please clarify.*

**Response:**

Motivated by this useful suggestion, we analyzed the precipitation in the forecast data and found that models tend to predict excess precipitation. The overestimated precipitation can explain why predicted snow cover increases with forecast lead times (Fig. 6 in the revised manuscript).

"Studies have shown that current state-of-the-art atmospheric general circulation models (GCMs) tend to strongly overestimate the precipitation over the Tibetan Plateau (e.g., Su et al., 2013; Chen and Frauenfeld, 2014; Zhang et al., 2016; Zhang et al., 2019). For example, Su et al. (2013) evaluated 24 GCMs that were available in the fifth phase of the Coupled Model Intercomparison Project (CMIP5) over the eastern Tibetan Plateau by comparing the model outputs with ground observations, and they found that all of the models consistently overestimated the observed precipitation for all seasons. Zhang et al., (2019) found similar results, in that all climate models they evaluated exaggerated the daily precipitation in the Tibetan Plateau during winter compared with the observed values. Here, we also found that the S2S models tended to overestimate the precipitation over the Tibetan Plateau. We compared the precipitation in the S2S models with both the gauges-based GPCC precipitation dataset and the satellite-based TRMM precipitation dataset (Fig. 6). The regional averaging wintertime mean precipitation over the Tibetan Plateau in the GPCC and TRMM models are 0.27 mm day$^{-1}$ and 0.32 mm day$^{-1}$, respectively. Compared with the overserved precipitation, all three S2S models exaggerate the regional precipitation obviously. Notably, such exaggerations always exist throughout the model integration. The ECMWF model reproduces the precipitation that is closest to the observations among the three models, but it still shows a large overestimation. The precipitation in the ECMWF model is 0.78 mm day$^{-1}$ to 0.88 mm day$^{-1}$. The precipitation values in the NCEP model (1.07 mm day$^{-1}$ to 1.37 mm day$^{-1}$) and in the CMA model (1.50 mm day$^{-1}$ to 2.13 mm day$^{-1}$) have larger precipitation biases and even increase with the forecasting lead time. These overestimations of the precipitation induce underestimations of the TPSC dissipation, and they lead to positive biases in the TPSC from the models. Because the overestimation of the precipitation exists throughout the model integration, the positive biases of the TPSC accumulate and increase with the model integration." (in the revised Section 3)

"All of the three S2S models consistently exaggerate the precipitation over the Tibetan Plateau compared to the observations. The exaggeration of the precipitation is prominent and always exists throughout the model integration. Systematic bias in the TPSC therefore occurs and accumulates with the model integration time due to exaggeration of the precipitation in the models." (in the revised Section 5)

[Figure]

**Figure 6 in the revised manuscript.** The multiyear wintertime mean precipitation over the Tibetan Plateau (unit: mm day$^{-1}$) for the observations and forecasts in each model.

*L244: how is the GDAS snow analysis used in Section 4.3 on numerical modelling compare with the IMS snow analysis, used in the first part of the paper. While it is mentioned that GDAS assimilates IMS, does it assimilate it over the TP? What does it assimilate specifically over the TP (in-situ data?)? Would the prediction skill be different if evaluated against GDAS (Fig 1) ?*

**Response:**

The main conclusion in this study is that there is a systematic positive bias in the TPSC, and this bias increases with forecast lead times. To support this conclusion, the multiyear winter mean TPSC derived from the S2S models are compared to the observations.

Following your suggestion, we checked the multiyear winter mean TPSC in the GDAS (the FNL analyses). The multiyear winter mean TPSC index is 28.5% in the FNL analyses, and 31.9% in the IMS analyses. Compared to the systematic bias of TPSC in S2S models presented in this study, the difference of winter mean TPSC index between the FNL and the IMS are much smaller and do not change the main conclusion.

*Conclusions: a brief mention of possible, relevant physical processes over the TP leading to snow ablation would be helpful. Could it be the strong surface winds or else the snow sublimation missing in the models? The short length of the period over which the forecasts are evaluated (around 10 years) is a bit of a concern. It appears that the biases are quite strong and systematic, but I wonder if some features in the forecasts would be robust over a longer period: for example, the slowdown in TPSC in early winter (December) seen in ECMWF and NCEP (Fig 3). I realise that if adding another 10 years may entail a lot of computational work, but it would add to the robustness of the conclusions. At least, a word of caution in the summary is warranted.*

**Response:**

1. In the conclusion, we ascribed the sources of the systematic bias of TPSC prediction could come from the biased precipitation in the models. We agree with the reviewer that the

surface winds and snow sublimation could also play a role in causing the snow ablation. Identifying the relative contributions of these factors to the biased snow prediction needs more detailed and careful diagnoses. A discussion was added in Section 5. "Surface winds and snow sublimation could also play a role in causing the snow ablation. Identifying the relative contributions of these factors to the biased snow prediction needs more detailed and careful diagnoses. Future studies on this issue are potentially valuable." We will address this issue in the near future. Thank you for your comments.

2. We conducted another 10 years of numerical experiments. The experiments in the revised manuscript ran for 20 winters (from 2000/2001 to 2019/2020).

*Typos / English*
*L43: hydrologic cycle*
*L28: radiative rather than radiant*
*L108: the total variability*
*L113: and the three different*
*L143: the preceding week rather than the last week, seems more appropriate*
*L152: accumulation, leading to a systematic TPSC bias.*
*L168: growth is used for a declining variable. Either decrement, reduction or decline should be clearer.*
*L173: real rate should be observed rate or rate derived from IMS.*
*L173: indices or index*
*L202: growth of SAT, rather than TPSC (it says 1.2 degree).*
*L229: land-atmosphere interactions*
*L256: Hence,*

**Response:**

All of these are corrected. We thank the reviewer for your careful reading of the manuscript.

---

## Author Response (AR3)

Dear Reviewer,

We would like to thank you again for your careful reading of our manuscript and helpful suggestions.

The following is our point-by-point response. The reviewer's comments are shown in **_blue italics_**. Our responses are provided in **black**. The revised text is in **red**.

Sincerely,
All of the authors
* * *
*Reviewer #1*

*Re-review of "Systematic bias of Tibetan Plateau snow cover in subseasonal-to-seasonal models", by Wenkai Li, Shuzhen Hu, Pang-chi Hsu, Weidong Guo, and Jiangfeng Wei, submitted to The Cryosphere.*

*The authors incorporated the comments from the first round of review and the manuscript has much improved, but some drawbacks below still need to be addressed. I recommend a minor revision prior to publication of this manuscript in The Cryosphere.*

*Specific comments:*

*1. The numerical experiments in section 4.3 is emphasizing the relationship between the cold SAT biases and the overestimation of TPSC. However, this part is not mentioned in the abstract at all, which is suggested to be included.*

**Response:**

Many thanks for your comment. Following your suggestion, a brief summary of the numerical experiments were added in the revised manuscript.

"Numerical experiments further confirm the causality. These incorrect cooling shifts over the Tibetan Plateau are caused by the systematic biases of TPSC." (in the revised abstract)

*2. Please give detailed descriptions of the experimental design in Section 2.3 such as the horizontal resolution, the option of nesting, the method of significant test et al.*

**Response:**

Agree. These are important information that should be added.

"The simulation domain is in a cylindrical Equidistant projection with a horizontal resolution of 1°×1° and located within 5–65°N and 40–170°E (as shown in Fig. 1) and without nesting. There are 41 levels in the vertical direction."

"We averaged all 340 cases in CTL runs and EXP runs respectively. Ensemble mean results between CTL and EXP are compared with each other." (in the revised Section 2.3)

*3. The increasing upward-reflected shortwave correponds to a negative value in Figure 11c. The positive direction in the diagnosis of surface energy balance needs to be clarified to avoid misunderstanding.*

**Response:**

Agree. To avoid confusion, we have clarified the signs of surface fluxes in the revised caption of Figure 11. Figure 11c and related text (the last paragraph in Section 4.3) have also been revised accordingly.

"The flux sign is positive downwards." (in the revised caption of Figure 11)

*4. What is the performance of the surface energy budget in S2S models? Can the relationship of TPSC and SAT be interpreted in S2S models just as the numerical experiments? Please clarify*

**Response:**

Excellent comment. To address this question, we diagnosed the surface energy budget using the predictions of S2S models. Here the differences in surface energy fluxes over the Tibetan Plateau between forecasts with a lead of 4 weeks and forecasts with a lead of 1 week in each S2S model are shown (see following Fig. R1). The differences contain responses of surface energy fluxes to biases of TPSC (as shown in Fig. 3b and 4).

The surface energy budget in S2S models and that in our WRF experiments show some similarities. The upward-reflected shortwave radiation (↑SW) increases with forecast lead time (negative value indicates enhanced upward radiation). Thus, the net shortwave radiation (NetSW) is decreased. The sensible heat flux (SH) also decreases with forecast lead time (positive values indicates reduced upward SH). The changes in ↑SW, NetSW and SH are consistent with WRF experiments. However, the changes in these three terms seem to be not as large (or evident) as the WRF experiments. Another evident difference between S2S models and WRF experiments is the long wave radiation. In WRF experiments, the response of the net longwave radiation (NetLW) is much smaller than that of the NetSW. In S2S models, the changes of NetLW are as evident as

that of NetSW. These differences between WRF experiments and S2S models could be related to the complex land-air interaction in the S2S models (rather than the one-way forcing of TPSC for WRF experiments). The surface energy budget in S2S models contain signals not only from the TPSC-forced response but also from the internal atmospheric variability contributing to the surface energy variability. The differences in land surface scheme used in these models may also be the factor leading to distinct results. We will leave these issues for our future study. Thank you for raising this important and interesting question.

[Figure]

**Figure R1**. Differences in the multiyear wintertime mean surface energy fluxes over the Tibetan Plateau between forecasts with a lead of 4 weeks and forecasts with a lead of 1 week in each S2S model. The terms from left to right are downward shortwave radiation (↓SW), downward longwave radiation (↓LW), upward shortwave radiation (↑SW), upward longwave radiation (↑LW), net shortwave radiation (NetSW), net longwave radiation (NetLW), sensible heat flux (SH), latent heat flux (LH) and ground heat flux (GRD) at the surface over the Tibetan Plateau, respectively (unit: W m$^{-2}$). The flux sign is positive downwards.

*5. Page 13 Line 412: a real rate -> an observed rate*

**Response:**

We revised the word according to your suggestion.

*6. Figure 5: the x-axis label "Time(day)" is not agreement with the figure caption, please check.*

**Response:**

The labels are corrected. Many thanks for your careful reading of the manuscript.
* * *
Dear Reviewer,

We would like to thank you again for your careful reading of our manuscript and helpful suggestions.

The following is our point-by-point response. The reviewer's comments are shown in **blue italics**. Our responses are provided in **black**. The revised text is in **red**.

Sincerely,
All of the authors
* * *
*Reviewer #2*

*The authors have addressed most of my comments, and substantially revised and improved the manuscript.*

*I believe that there may be an incorrect statement that possibly needs to be changed. I hence recommend a minor revision.*

*1) Concerning the initialization of snow in the ECWMF system, the authors mention: "The snow mass is initialized by the ECMWF Land Data Assimilation System (LDAS). The snow initialization relies on a land surface synoptic report and national ground observations of the snow depth available on the Global Telecommunications System (GTS), as well as on the Interactive Multisensor Snow and Ice Mapping System (IMS) snow cover information."*

*I believe that this is correct for the operational real-time S2S forecasts, but the authors seem to use the reforecasts (stated L74). In the reforecasts, a re-analysis is used for initialization of the land states.*

*Taking from the webpage cited by the authors "If the initialization of the LSM for re-forecasts deviates from the procedure for forecasts, please describe the differences. The re-forecasts initialization is based on ERA-Interim and ERA-Interim/Land datasets, while the real-time forecasts are based on the IFS operational initial conditions of the ENS/EDA systems." It is unclear to me if the re-analysis that is used now, is based on the ERA5 system or still ERA-Interim.*

*Please clarify these points if necessary, for the sake of correctness (although this will not change the results).*

*In any case, note that the ECMWF LDAS would not use IMS in high altitude regions, such as the Tibet, nor incorporate in-situ snow depth measurements in that region.*

**Response:**

Indeed, this point is important. According to your comment, we checked the S2S model description (https://confluence.ecmwf.int/display/S2S/Models) again and other references (e.g., Son et al., 2020). Exactly as you said that the operational real-time S2S forecasts in ECMWF and the reforecast in ECMWF uses different land initializations. "The reforecasts initialization is based on ERA-Interim and ERA-Interim/Land datasets, while the real-time forecasts are based on the IFS operational initial conditions of the ENS/EDA systems." We have revised the description in our manuscript.

"For the ECMWF model, the reforecasts initialization is based on ERA-Interim and ERA-Interim/Land datasets. The daily Interactive Multisensor Snow and Ice Mapping System (IMS) snow-cover product has been used to constrain the ERA-Interim snow analysis (Dee et al., 2011)." (in revised Section 2.1)

*Suggestions for wording*
*L199: occur at a regional scale*

**Response:**

This has been modified as suggested.

*L247: such an overestimation persists throughout the model integration*

**Response:**

This has been modified as suggested.

*L294: I believe you mean the \*decline of the SAT\*, not the growth (Fig 10b) (?)*

**Response:**

Yes. This has been corrected.

*L306: this sentence is a repeat of line L286 in the same section.*

**Response:**

This sentence has been deleted. We thank the reviewer again for your careful reading of the manuscript.
* * *

[revised manuscript text omitted]

---

## Author Response (AR4)

Dear Editor,

Thank you very much for handling our manuscript. The followings are response to your comments.

*1) You provided a thorough response to comment #4 from Reviewer 1 (What is the performance of the surface energy budget in S2S models? Can the relationship of TPSC and SAT be interpreted in S2S models just as the numerical experiments? Please clarify), but it does not appear that you included any modifications to the manuscript that reflect this discussion. Do you want to incorporate any of this in the final manuscript?*

The surface energy budgets in S2S models, as shown and discussed in last response, support the conclusions in our manuscript. But the energy budgets in S2S models contain complex land-air interaction (rather than the one-way forcing of TPSC for WRF experiments). To make the results concise, we prefer not to add this picture to the current manuscript. Instead, we would like to see study on the surface energy budgets in S2S models in future separate work.

*2) The modification to the abstract, suggested by Reviewer 1, seems appropriate to me, but the end of the abstract now seems redundant, as a sentence prior to the addition reads "Such systematic biases of TPSC influence the forecasted surface air temperature in the S2S models." You can edit the abstract if you want, or leave it as is. Your call.*

The last sentence has been deleted. The sentence "Numerical experiments further confirm the causality" is retained.

We hope the revised manuscript are acceptable, and look forward to hearing from you with a decision regarding our paper. Thank you.

Sincerely,
All of the authors

[revised manuscript text omitted]